# RoboFace: Face Restoration Made Robust via Implicit and Explicit Textual Guidance

## Abstract

Existing blind face restoration methods often struggle with out-of-distribution degradations. While high-level latent spaces like discrete codebooks offer some robustness, they frequently introduce unnatural artifacts under severe corruption; meanwhile, alternative approaches depend on costly degradation scaling and large-scale diffusion model retraining. In this paper, we propose **RoboFace**, a novel framework that achieves robust face restoration by prompting a pre-trained diffusion model with dual textual guidance. Given low-quality inputs, Robo-Face constructs a structured, semantic-aligned space through two complementary guides: *implicit guidance* from CLIP latent features to preserve visual fidelity and identity, and *explicit guidance* from natural text prompts for diffusion prior and flexible, user-interactive control. These guides are seamlessly integrated via a thoughtfully designed Decoupled Cross-Attention (DCA) module, which adaptively aligns them with the pretrained diffusion model. Extensive experiments demonstrate that RoboFace is exceptionally robust across a wide spectrum of degradations, delivering state-of-the-art results even on challenging low-quality surveillance faces. Our results highlight the promise of semantic guidance as a reliable and flexible paradigm for robust face restoration.

## 1 Introduction

Blind face restoration (BFR) seeks to recover high-quality (HQ), photorealistic facial images from low-quality (LQ) inputs degraded by a complex and unknown process. This capability is crucial for a wide range of real-world applications such as enhancing legacy photos, restoring compressed social media images, and improving low-resolution surveillance. However, BFR faces two fundamental challenges: 1) the difficulty of accounting for the diverse and out-of-distribution (OOD) degradation patterns encountered in real-world scenarios, and 2) its inherently ill-posed nature, where a single LQ image may correspond to multiple plausible HQ counterparts. These challenges are often entangled, making it extremely difficult to design a robust BFR model.

To address these challenges, existing methods have primarily adopted two strategies, both with significant limitations. The first aims to better mimic real-world corruptions, as in Real-ESRGAN (Wang et al., 2021b), which introduced a second-order degradation process. Yet, any synthetic design inevitably suffers from distribution mismatch with the vast, unpredictable space of real-world degradations. Simply increasing the complexity of the degradation model is not a scalable solution; as the input space expands, the restoration problem becomes increasingly ill-posed. The second leverages powerful priors to constrain the solution space. However, codebook-based priors, used in methods like CodeFormer (Zhou et al., 2022), suffer from ambiguity where a single code can represent both clean and degraded patches (Fig. 1(a)). While subsequent work like DAEFR (Tsai et al., 2024) mitigates this with a separate LQ encoder, the unconstrained nature of the learned codebook still compromises robustness against diverse real-world degradations (see results in Fig. 1). More recently, pre-trained diffusion models have been employed as generative priors to enforce realistic outputs (Yue & Loy, 2024). Yet, these methods often treat the LQ input as a weak condition (Fig. 1(b)), which proves effective for simple degradations but falters when faced with complex, severe corruptions, as shown in Fig. 1.

In this paper, we introduce a novel framework, **RoboFace**, that re-frames the restoration task by operating within a *semantic-aligned structured space* that robustly bridges the LQ and HQ domains.

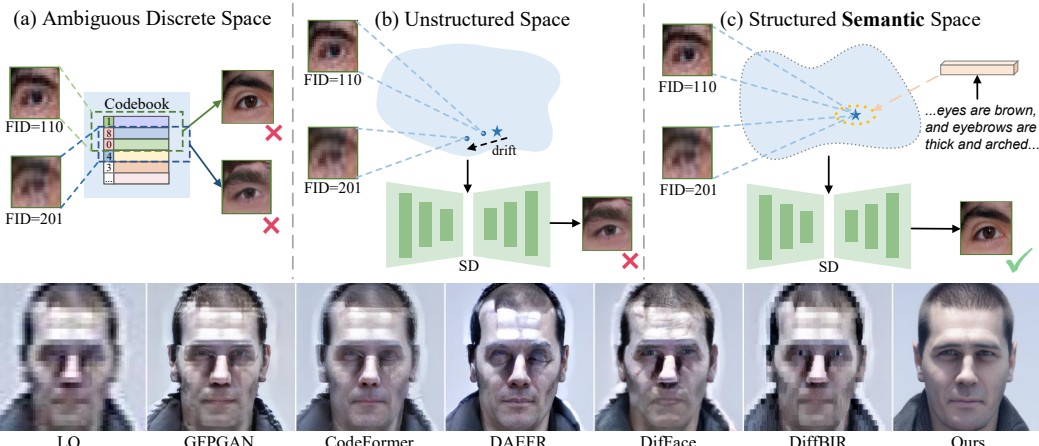

Figure 1: Motivation of our method. (a) The codebook latent space is ambiguous; (b) using LQ as a weak condition for the diffusion prior is vulnerable to OOD degradations; (c) the proposed structured semantic space is both robust and effective. Example result on SCface (Grgic et al., 2011) show that our approach achieves superior robustness under real-world surveillance image.

This space is aligned with high-level textual semantics, which provides a powerful intermediate representation as shown in Fig. 1(c). For the LQ input, a textual representation is naturally robust to diverse degradation patterns. Our linguistic system abstracts the underlying concepts (*e.g.*, "a smiling woman") and remains largely invariant to degradations that are difficult to describe explicitly. For the HQ output, pre-trained text-to-image diffusion models have demonstrated an exceptional ability to transform these high-level textual features into a photorealistic image space. With this textual bridge, RoboFace circumvents the need to explicitly model the complex degradation space, instead focusing on translating the invariant semantic essence of the LQ input into a high-fidelity reconstruction.

To realize this, RoboFace prompts a pre-trained diffusion model using dual textual guidance: *implicit guidance* from CLIP latent features to preserve visual fidelity and identity, and *explicit guidance* from natural language text prompts for diffusion prior and flexible, user-interactive control. These two guides are seamlessly integrated via a thoughtfully designed Decoupled Cross-Attention (DCA) module. This module sequentially adapts the two guides to the pretrained diffusion model through cross-attention in an explicit-then-implicit order. This strategy is highly effective, as it first leverages the explicit text to set the overall semantic context, preserving the powerful generative capabilities of the text-to-image model, and then integrates the implicit visual clues to ensure the final output is faithful to the original identity.

Through this design, RoboFace constructs a semantic-aligned latent space that is both robust to a wide spectrum of degradations and adaptable to user intentions. Our main contributions are summarized as follows:

- We propose **RoboFace**, a novel framework for blind face restoration that utilizes a dual textual guidance paradigm to robustly navigate the space between degraded inputs and high-quality outputs, bypassing the need to model complex degradations.
- We introduce a **Decoupled Cross-Attention (DCA)** module that effectively integrates implicit visual guidance and explicit text guidance, maximizing fidelity while retaining the strong generative prior of pre-trained diffusion models.
- We demonstrate through extensive experiments that RoboFace achieves exceptional robustness against severe, out-of-distribution degradations where previous methods often fail.

## 2 RELATED WORK

### 2.1 BLIND FACE RESTORATION

Early blind face restoration (BFR) methods leverage *StyleGAN priors*, where approaches like GF-PGAN (Wang et al., 2021a) and GPEN (Yang et al., 2021) encode degraded inputs into the Style-

GAN latent space (Karras et al., 2019) to achieve photorealistic reconstructions. However, these methods often fail under complex real-world degradations, producing artifacts or distorted identities. To improve robustness, *codebook-based priors* such as CodeFormer (Zhou et al., 2022), RestoreFormer (Wang et al., 2022), and DAEFR (Tsai et al., 2024) exploit vector-quantized dictionaries (Esser et al., 2021; Van Den Oord et al., 2017) for more faithful and controllable results. However, the discrete codebook introduces ambiguity, as clean and degraded patches may map to the same code, leading to unnatural textures and identity mismatches under severe degradations. Recently, *diffusion-based priors* have advanced BFR by framing it as conditional generation (Wang et al., 2023b; Yu et al., 2024; Yue & Loy, 2024; Lin et al., 2024b; Chen et al., 2025). DifFace (Yue & Loy, 2024) maps LQ inputs into intermediate denoising states, while DR2 (Wang et al., 2023b) and DiffBIR (Lin et al., 2024b) adopt a two-stage paradigm of degradation removal followed by refinement. Despite their realism, these methods still treat the LQ input as a weak condition, making them unreliable against severe noise, blur, or compression.

## 2.2 DIFFUSION-BASED NATURAL IMAGE RESTORATION

Given the strong generative capabilities of diffusion models in producing realistic images, recent studies have increasingly adopted diffusion-based approaches for natural image restoration. Several methods further exploit additional modalities to guide the restoration process. For instance, SSP-IR (Zhang et al., 2025) encodes textual descriptions into feature embeddings and integrates them with image features to enhance restoration performance. TextualDegRemoval (Lin et al., 2024a) denoises in the textual space to obtain cleaner guide images, while DA-CLIP (Luo et al., 2024) leverages text supervision to disentangle clean image content from degradation noise in the latent space. DPIR (Kong et al., 2025) introduces a degradation-robust encoder–decoder to generate clean visual representations and further incorporates textual features for guidance. Despite these advances, challenges remain in both generalization and restoration quality. Existing approaches often rely on visual representations to recover local details or employ broad textual descriptions for auxiliary guidance. In contrast, our method leverages global and semantically rich textual guidance to capture holistic semantics and identity cues, while also exploiting the inherent generalization capacity of the textual space. This design leads to superior robustness against diverse and unpredictable degradations.

## 3 METHOD

### 3.1 OVERVIEW

The proposed RoboFace leverages both explicit and implicit textual guidance to handle out-of-distribution degradations and enhance visual quality. To obtain a clean implicit text representation, we first train the implicit text extractor to capture informative facial identity features while filtering out degradations in the high-level semantic space (Fig. 2(a)). Meanwhile, the explicit text extractor generates face-specific descriptive prompts. The explicit and implicit representations are integrated through a Decoupled Cross Attention mechanism (Fig. 2(c)). Furthermore, to supplement the restoration with crucial image details that semantic information cannot adequately provide, we incorporate ControlNet (Zhang et al., 2023). RoboFace robustly addresses complex degradations and produces visually appealing restoration results.

### 3.2 STAGE I: LEARNING IMPLICIT TEXTUAL GUIDANCE

A robust proxy space is essential for handling OOD degradations. We select the textual embedding space of diffusion models for this purpose, as it is inherently aligned with photorealistic, high-quality images and is naturally invariant to non-semantic degradation patterns that are difficult to describe. However, natural language has a limited representational capacity; explicitly describing a LQ input and using that text as guidance can lead to information loss. We therefore propose to learn a more informative, high-level proxy representation *implicitly* from the LQ inputs.

The process begins with the Mapper $\phi_{\text{mapper}}$ which projects $\mathbf{F}_{\text{clip}}$ from a frozen CLIP image encoder into the textual conditioning space of pretrained diffusion model. This initial representation, $\mathbf{F}_{\text{mapper}}$, captures high-level semantics as a sequence of $N = 30$ learnable tokens:

$$\mathbf{F}_{\text{mapper}} = \phi_{\text{mapper}}(\phi_{\text{clip-img}}(\mathbf{I})), \qquad \mathbf{F}_{\text{mapper}} \in \mathbb{R}^{N \times D}. \tag{1}$$

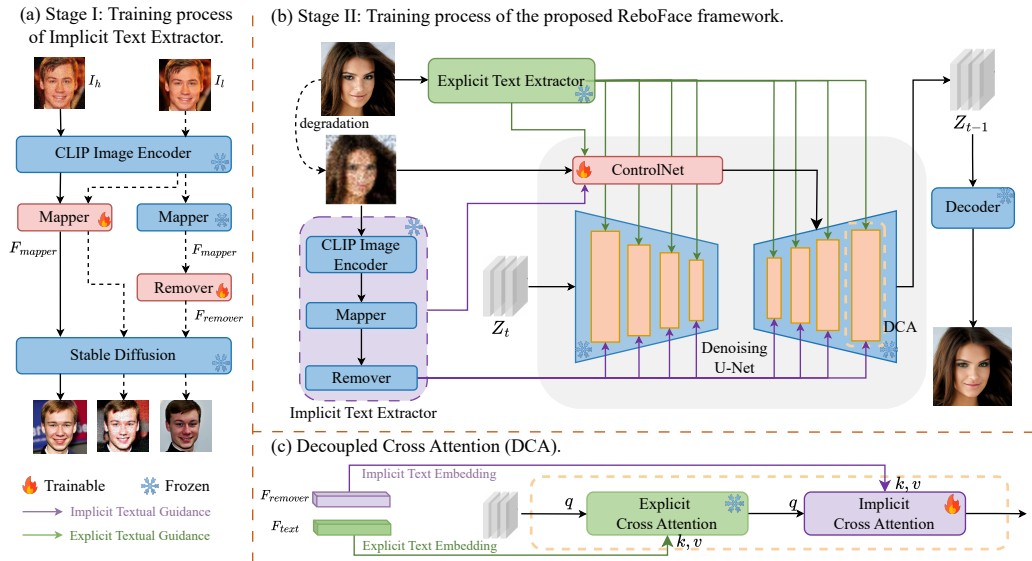

Figure 2: Framework of our proposed RoboFace. Given an LQ input under extreme degradation, two complementary guidance signals, implicit and explicit text extractors, are collaboratively aligned with a pretrained diffusion model via the proposed decoupled cross-attention. During training, *HQ images* are used to generate explicit text embeddings, while in inference, the *remover outputs* are leveraged to produce high-fidelity text embeddings.

While effective, $\mathbf{F}_{\text{mapper}}$ still retains corruption-related artifacts when generated from an LQ image. Therefore, a second network, the Remover $\phi_{\text{remover}}$, is tasked with purifying this representation to produce the final clean feature, $\mathbf{F}_{\text{remover}}$:

$$\mathbf{F}_{\text{remover}} = \phi_{\text{remover}}(\mathbf{F}_{\text{mapper}}), \qquad \mathbf{F}_{\text{remover}} \in \mathbb{R}^{N \times D}. \tag{2}$$

The training for this stage proceeds sequentially, optimizing both $\phi_{\text{mapper}}$ and $\phi_{\text{remover}}$ with the standard diffusion loss while *keeping all pre-trained components frozen*:

$$\mathcal{L}_{\text{stage I}} = \mathbb{E}_{z_t, \epsilon \sim \mathcal{N}(0, I), t} \left[ \| \epsilon - \epsilon_\theta(z_t, t, \mathbf{F}) \|_2^2 \right], \tag{3}$$

where $\mathbf{F}$ is $\mathbf{F}_{\text{mapper}}$ or $\mathbf{F}_{\text{remover}}$. First, the $\phi_{\text{mapper}}$ is trained with an autoencoding objective on both LQ and HQ images, ensuring it learns a general-purpose visual-to-textual projection. With $\phi_{\text{mapper}}$ frozen, the Remover is then trained exclusively on LQ-HQ pairs $\{(\mathbf{I}_l, \mathbf{I}_h)\}$. Its sole objective is to take the noisy feature derived from $\mathbf{I}_l$ and denoise it into a clean representation, $\mathbf{F}_{\text{remover}}$, that can successfully guide the reconstruction of the ground-truth image $\mathbf{I}_h$.

## 3.3 STAGE II: FUSING GUIDANCE VIA DECOUPLED CROSS-ATTENTION

This stage combines the implicit guidance from Stage I with new, explicit textual guidance using a cross attention mechanism to achieve robust and high-fidelity face restoration.

**Generating Explicit Textual Guidance.** The role of explicit guidance is to provide high-level semantic cues that establish a global context for the restoration. During training, we generate these explicit prompts by applying LLaVA (Liu et al., 2023a) to the ground-truth HQ images to extract detailed, face-specific descriptions.

At inference, generating reliable prompts directly from LQ inputs is challenging. We solve this by first using our trained Stage I model (Sec. 3.2) to produce a clean, intermediate restoration. We then apply LLaVA to this intermediate image to extract the explicit prompt. This strategy ensures the explicit guidance remains robust and faithful to the original identity, even under severe degradation, providing a meaningful semantic anchor for the final restoration.

**Decoupled Cross-Attention (DCA).** To effectively fuse the two guidance signals without mutual interference, we introduce the Decoupled Cross-Attention (DCA) module. Instead of naively com-

bining the features, DCA enforces a sequential, coarse-to-fine refinement process. First, the explicit text prompt $\mathbf{F}_{\text{text}}$ is injected to establish the global semantic context, narrowing the solution space by leveraging the powerful priors of the pre-trained diffusion model. Subsequently, the implicit guidance $\mathbf{F}_{\text{remover}}$, which carries rich visual cues from the input image, is incorporated to refine identity consistency and other fine-grained details. This explicit-then-implicit ordering allows DCA to maximize fidelity and robustness simultaneously.

**Training Objective.** With the weights of $\phi_{\text{mapper}}$ and $\phi_{\text{remover}}$ frozen, the main blind face restoration model is trained end-to-end. At each denoising step, the implicit guidance $\mathbf{F}_{\text{remover}}$ and the explicit guidance $\mathbf{F}_{\text{text}}$ are fed into the diffusion U-Net at multiple scales via our DCA module. The model is optimized using the standard noise prediction loss:

$$\mathcal{L}_{\text{stage II}} = \mathbb{E}_{z_t, \epsilon \sim \mathcal{N}(0, I), t} \left[ \| \epsilon - \epsilon_\theta(z_t, t, \text{DCA}(\mathbf{F}_{\text{text}}, \mathbf{F}_{\text{remover}})) \|_2^2 \right], \tag{4}$$

where $\text{DCA}(\cdot)$ represents the conditioning produced by our decoupled attention mechanism.

# 4 EXPERIMENTS

## 4.1 EXPERIMENTAL SETTINGS

**Training Datasets.** We train all models on the FFHQ dataset (Karras et al., 2019), which contains 70,000 high-quality face images resized to $512 \times 512$. To create the LQ counterparts for training, we adopt a second-order degradation process (Wang et al., 2021b) and the specific parameter are detailed in the first row of Tab. 1 (see Appendix Sec. C.1 for details). To obtain explicit text guidance, we use LLaVA (Liu et al., 2023a) to generate face description prompts for each image.

**Testing Datasets.** We follow the evaluation protocol of CodeFormer (Zhou et al., 2022) and assess our method on one synthetic dataset, CelebA-Test, and three real-world datasets: LFW-Test (Huang et al., 2008) (1,711 images with mild degradations), WIDER-Test (Yang et al., 2016) (970 images with severe degradations), and SCface (Grgic et al., 2011). To evaluate OOD performance, we replace WebPhoto-Test (medium degradation) with SCface, which presents more challenging surveillance degradations, including aliasing, jagged artifacts, low-light noise, and defocus blur. The SCface dataset comprises 130 subjects recorded by five different surveillance cameras; for evaluation, we use images captured at a distance of 2.6 meters.

**Evaluation Metrics.** We adopt both reference-based and no-reference image quality metrics for comprehensive evaluation. For synthesized datasets with ground truth, we use FID (Bynagari, 2019), LPIPS (Zhang et al., 2018), as well as no-reference perceptual metrics MANIQA (Yang et al., 2022) and CLIPIQA (Wang et al., 2023a) to better capture perceptual quality. For real-world datasets without ground truth, we evaluate performance using FID and CLIPIQA.

**Implementation Details.** Both the Mapper and Remover are configured with 30 tokens and trained for one epoch on the FFHQ dataset during Stage I, using a batch size of 4, the Adam optimizer (Kingma & Ba, 2015) and a learning rate of $1 \times 10^{-6}$ on a single NVIDIA V100 GPU. Our RoboFace model is built upon the pretrained stable-diffusion-2.1 and trained for 15 epochs on two NVIDIA RTX 4090 GPUs, with a batch size of 192, using Adam and a learning rate of $5 \times 10^{-5}$.

## 4.2 COMPARISON WITH THE STATE-OF-THE-ART METHODS

We compare RoboFace with seven state-of-the-art BFR methods across three categories: (1) StyleGAN-prior methods: GFPGAN (Wang et al., 2021a) and GPEN (Yang et al., 2021); (2) Codebook-prior methods: CodeFormer (Zhou et al., 2022) and DAEFR (Tsai et al., 2024); (3) Diffusion-based methods: DR2 (Wang et al., 2023b), DifFace (Yue & Loy, 2024) and DiffBIR (Lin et al., 2024b). All experiments are conducted using official code.

**Robustness Comparison.** We rigorously evaluated the ability of RoboFace to generalize to OOD degradations by creating three CelebA-Test splits with progressively severe artifacts. These splits correspond to degradations that are in-distribution (FID=110.50), partially OOD (FID=199.68), and fully OOD (FID=262.42), with the last containing degradation patterns entirely unseen during training. The hyper-parameters are provided in Tab. 1.

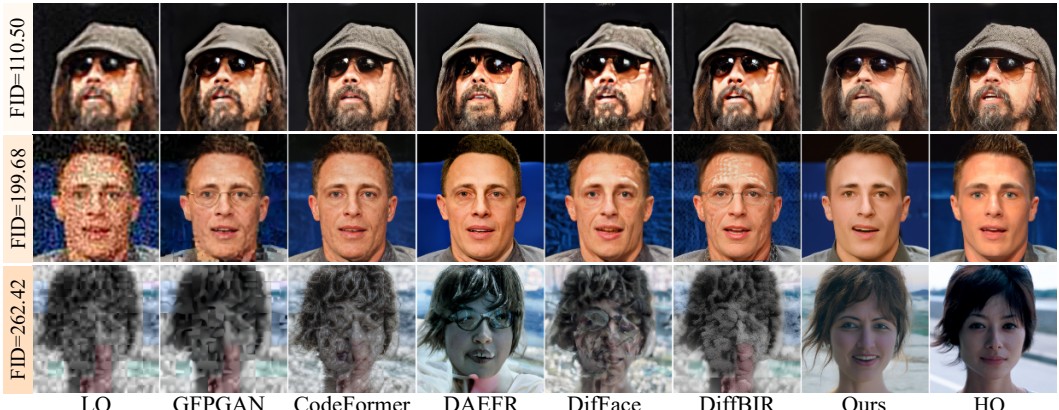

Figure 3: Qualitative comparisons on the synthetic CelebA-Test dataset under three degradation levels. Trained with the least degraded data, our method generalizes well to severe degradations and clearly outperforms the compared approaches.**(Zoom in for details)**

| FID Level | OOD | Gaussian Blur | | Downsample Factor | | Gaussian Noise | | JPEG Compression | |
|---|---|---|---|---|---|---|---|---|---|
| | | first order | second order | first order | second order | first order | second order | first order | second order |
| 110.50 (Train) | No | [0.2, 3.0] | [0.2, 1.5] | [0.15, 1.5] | [0.3, 1.2] | [1, 30] | [1, 25] | [30, 95] | [30, 95] |
| 199.68 (Test) | Partial | [1.2, 6.0] | [1.0, 4.0] | [0.02, 0.9] | [0.1, 0.8] | [15, 55] | [15, 40] | [5, 60] | [5, 60] |
| 262.42 (Test) | Yes | [6.0, 10.0] | [3.0, 6.0] | [0.05, 0.1] | [0.05, 0.2] | [60, 90] | [30, 45] | [1, 20] | [1, 20] |

Table 1: Degradation parameters with different FID levels on the synthetic CelebA-Test (second-order degradation strategy is applied, see Appendix Sec. C.1 for more details).

| FID | FID=110.50 | | | | FID=199.68 | | | | FID=262.42 | | | |
|---|---|---|---|---|---|---|---|---|---|---|---|---|
| Degradation | Within training range | | | | Overlap with training range | | | | Out of training range | | | |
| Methods | FID ↓ | LPIPS ↓ | MANIQA ↑ | CLIPIQA ↑ | FID ↓ | LPIPS ↓ | MANIQA ↑ | CLIPIQA ↑ | FID ↓ | LPIPS ↓ | MANIQA ↑ | CLIPIQA ↑ |
| GFPGAN | 17.03 | 0.29 | 0.45 | 0.57 | 91.05 | 0.50 | 0.36 | 0.53 | 257.77 | 0.75 | 0.40 | 0.41 |
| GPEN | 18.29 | 0.28 | 0.45 | 0.53 | 52.40 | 0.45 | 0.34 | 0.41 | 202.31 | 0.73 | 0.20 | 0.20 |
| CodeFormer | 14.26 | 0.27 | 0.52 | 0.68 | 41.54 | 0.40 | 0.48 | 0.68 | 132.04 | 0.60 | 0.39 | 0.64 |
| DAEFR | 14.18 | 0.27 | 0.54 | 0.68 | 21.72 | 0.36 | 0.53 | 0.67 | 82.40 | 0.56 | 0.43 | 0.65 |
| DR2 | 26.97 | 0.32 | 0.46 | 0.67 | 28.10 | 0.36 | 0.43 | 0.65 | 70.48 | 0.52 | 0.34 | 0.57 |
| DifFace | 14.93 | 0.30 | 0.45 | 0.56 | 41.67 | 0.41 | 0.31 | 0.54 | 152.06 | 0.61 | 0.22 | 0.36 |
| DiffBIR | 26.08 | 0.29 | 0.43 | 0.50 | 49.86 | 0.45 | 0.68 | 0.78 | 156.95 | 0.78 | 0.56 | 0.65 |
| **RoboFace** | 21.96 | 0.28 | 0.59 | 0.71 | 25.67 | 0.36 | 0.59 | 0.71 | 56.53 | 0.52 | 0.57 | 0.70 |

Table 2: Quantitative comparison on the synthetic CelebA-Test dataset under three degradation levels. The best results are marked in red, and the second best in blue.

The quantitative results in Tab. 2 reveal the limitations of existing methods, which decrease significantly on the OOD splits. In contrast, RoboFace demonstrates superior robustness, maintaining competitive FID and LPIPS scores while outperforming all competitors on all metrics in the fully OOD setting. These findings are supported by the qualitative examples in Fig. 3, which show that RoboFace preserves high visual fidelity and reconstructs natural facial details even when faced with complex and spatially variant noise. This highlights its exceptional robustness and capacity to handle challenging, real-world distribution shifts.

**General Real-world Datasets Comparison.** As shown in Tab. 3, our method achieves the best FID scores on LFW-Test and WIDER-Test, and consistently outperforms all competing methods on SCface, which involves extreme real-world surveillance degradations. While DiffBIR achieves relatively high CLIPIQA scores on the Mild and Heavy subsets, it fails on OOD dataset (SCface), producing corrupted or unstable results. In contrast, our method demonstrates strong robustness across diverse conditions, striking a good balance between perceptual quality and fidelity. Visual comparisons in Fig. 4 further confirm that our restorations remain stable under severe degradations and yield the most pleasing faces.

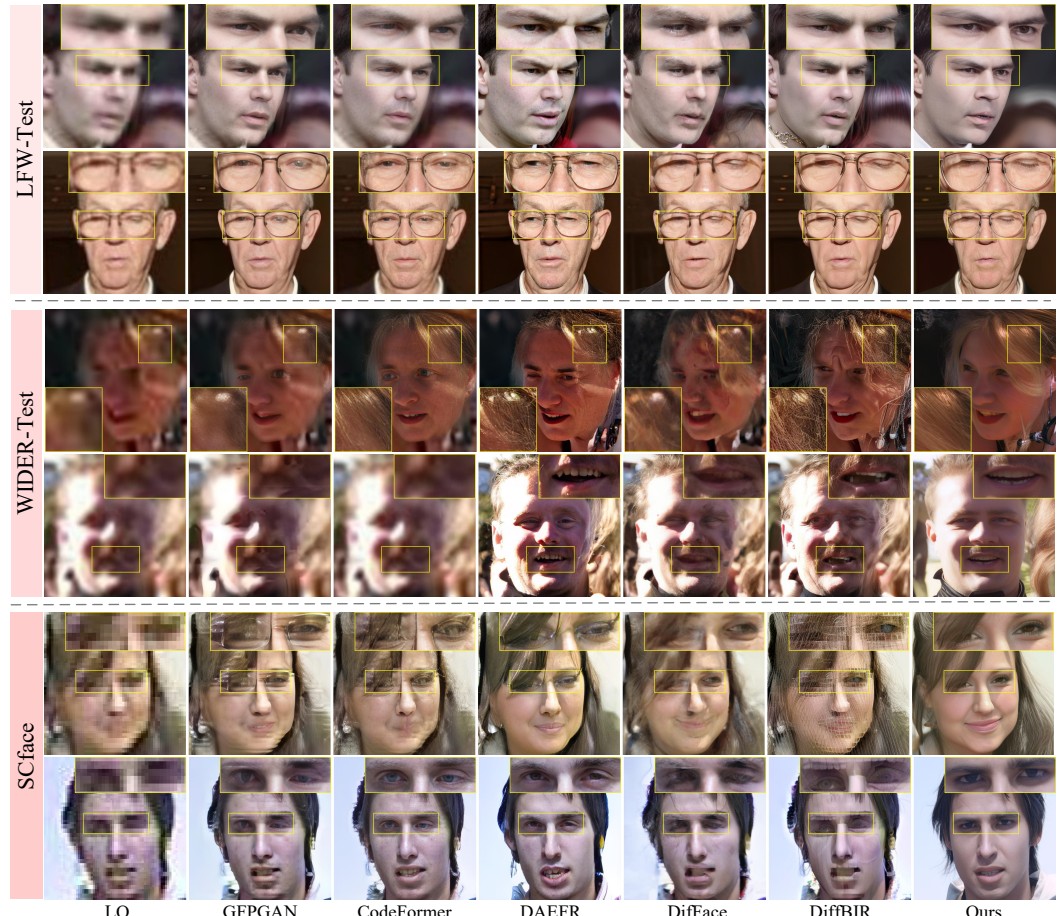

Figure 4: Qualitative comparison on real-world datasets of LFW-Test, WIDER-Test, and SCface. Our method is able to restore high quality faces, showing robustness to the heavy and out-of-distribution degradation.(**Zoom in for details**)

| Datasets | LFW-Test | | WIDER-Test | | SCface | | Average | |
| Degradation | Mild | | Heavy | | Surveillance | | | |
| Methods | FID ↓ | CLIPIQA ↑ | FID ↓ | CLIPIQA ↑ | FID ↓ | CLIPIQA ↑ | FID ↓ | CLIPIQA ↑ |
|---|---|---|---|---|---|---|---|---|
| GFPGAN | 53.87 | 0.63 | 50.36 | 0.58 | 106.97 | 0.65 | 70.40 | 0.62 |
| GPEN | 55.90 | 0.57 | 57.72 | 0.45 | 126.03 | 0.45 | 79.88 | 0.49 |
| CodeFormer | 52.84 | 0.69 | 39.22 | 0.70 | 99.07 | 0.69 | 63.71 | 0.69 |
| DAEFR | 47.69 | 0.70 | 36.72 | 0.70 | 103.64 | 0.72 | 62.68 | 0.71 |
| DR2 | 50.42 | 0.65 | 52.78 | 0.59 | 96.49 | 0.58 | 66.56 | 0.61 |
| DifFace | 46.67 | 0.61 | 37.70 | 0.59 | 106.34 | 0.45 | 63.57 | 0.55 |
| DiffBIR | 40.91 | 0.79 | 35.82 | 0.81 | 149.98 | 0.46 | 75.57 | 0.69 |
| **RoboFace** | 39.56 | 0.73 | 35.04 | 0.74 | 92.31 | 0.75 | 55.64 | 0.74 |

Table 3: Quantitative comparisons on real-world datasets of LFW-Test, WIDER-Test, and SCface. The best results are marked in red, and the second best in blue.

## 4.3 ANALYSIS OF ROBUSTNESS OF TWO TEXTUAL GUIDANCE

**t-SNE visualization for Implicit Textual Guidance.** To verify the efficacy of our textual-space denoising, we conducted a t-SNE analysis on feature embeddings from the CelebA-Test set. The visualization in Fig. 5 compares the raw CLIP features ($\mathbf{F}_{clip}$) with our denoised features ($\mathbf{F}_{remover}$). The results show two key improvements: 1) our method yields more compact clusters for each identity, and 2) it realigns degraded samples (both ID and OOD) with their high-quality counterparts. These effects confirm that our approach successfully strengthens identity-related semantics, leading to improved feature discriminability and robustness against unfamiliar degradations.

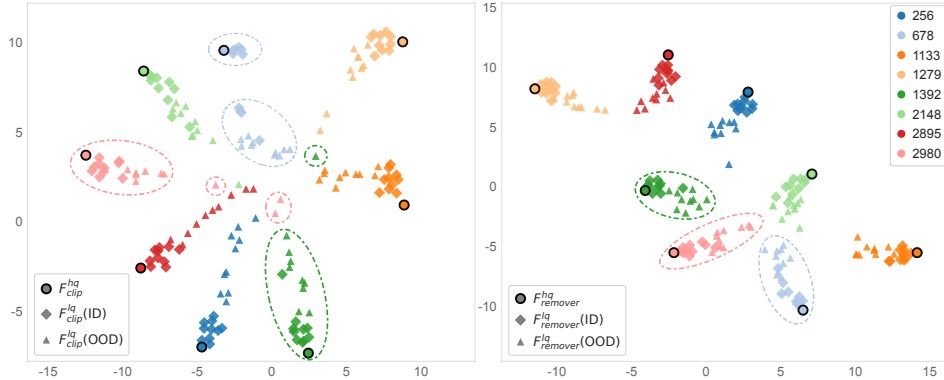

Figure 5: Visual comparison of CLIP features (left) and our refined features (right). Colors represent identities and markers denote data types. Raw CLIP features fragment identities into scattered subclusters, our method merges them into a single, compact cluster aligned with HQ samples.

**Text Similarity for Explicit Textual Guidance.**

For each LQ image, Stage I is applied to generate a denoised reference image. We then employ LLaVA (Liu et al., 2023a) to produce a caption for this reference and compute its similarity to the paired HQ caption using Qwen3 (Yang et al., 2025). For comparison, each LQ image is also restored using the baseline model GFPGAN (Wang et al., 2021a), cap-

| Dataset Text Similarity | CelebA-Test (FID=110.50) | CelebA-Test (FID=199.68) | CelebA-Test (FID=262.42) |
|---|---|---|---|
| $\text{sim}(\text{LQ}, \text{HQ})$ | 0.7686 | 0.7192 | 0.5903 |
| $\text{sim}_{\text{GFPGAN}}(\text{LQ}, \text{HQ})$ | 0.7700 | 0.7241 | 0.4875 |
| $\text{sim}_{\text{StageI}}(\text{LQ}, \text{HQ})$ | 0.7600 | 0.7349 | 0.6636 |

Table 4: Text-similarity comparison on CelebA-Test under different degradation levels.

tioned and evaluated in the same manner. As shown in Tab. 4, similarity between GFPGAN outputs and HQ captions decreases as degradation increases, whereas Stage I maintains higher similarity by denoising in the textual embedding space, demonstrating greater robustness.

## 4.4 ABLATION STUDIES

**Effects of Implicit and Explicit Textual Guidance.** We perform ablation studies to analyze the roles of implicit and explicit textual guidance in RoboFace. Incorporating implicit guidance suppresses noise and reduces artifacts, resulting in more faithful identity preservation (see Fig. 6). It also leads to higher-fidelity reconstructions, which is reflected by improved FID and LPIPS scores.

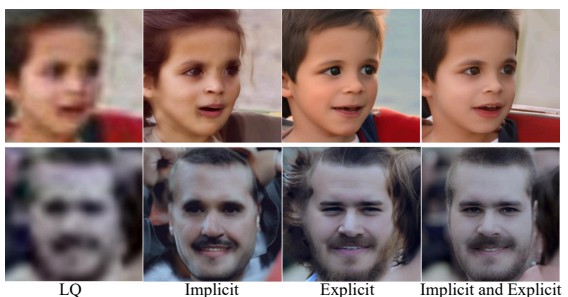

Figure 6: Qualitative comparison of the effectiveness of implicit and explicit textual guidance.

| Dataset | Metrics | w/o Explicit | w/o Implicit | Ours |
|---|---|---|---|---|
| WIDER-Test | FID ↓ | 35.41 | 37.49 | 35.04 |
| | MANIQA ↑ | 0.51 | 0.53 | 0.57 |
| | CLIPIQA ↑ | 0.70 | 0.71 | 0.74 |
| CelebA-Test (FID=110.50) | FID ↓ | 20.97 | 27.51 | 21.96 |
| | LPIPS ↓ | 0.34 | 0.32 | 0.28 |
| | MANIQA ↑ | 0.48 | 0.54 | 0.59 |
| | CLIPIQA ↑ | 0.74 | 0.70 | 0.71 |
| CelebA-Test (FID=262.42) | FID ↓ | 84.02 | 82.86 | 56.53 |
| | LPIPS ↓ | 0.57 | 0.60 | 0.52 |
| | MANIQA ↑ | 0.50 | 0.55 | 0.57 |
| | CLIPIQA ↑ | 0.68 | 0.70 | 0.70 |

Table 5: Quantitative comparison of implicit and explicit textual guidance.

On the other hand, explicit textual guidance demonstrates two key advantages. First, it enables high-level semantic control, allowing for an interactive and user-driven restoration process (see Fig. 7). Second, by grounding the restoration in the structured semantic space of pre-trained text-to-image models, this approach leverages their powerful priors. This foundation not only facilitates fine-grained control but also makes the process inherently more robust to degradation, resulting in superior perceptual quality, as confirmed by higher CLIPIQA and MANIQA scores in Tab. 5.

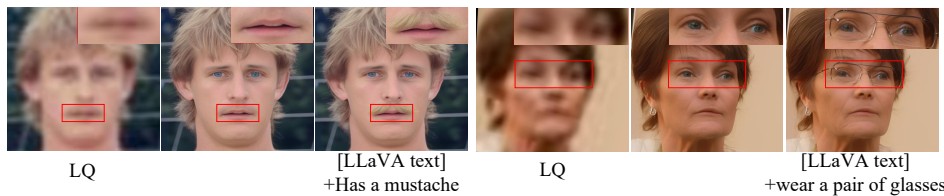

Figure 7: Visual results showing the controllability of explicit guidance.

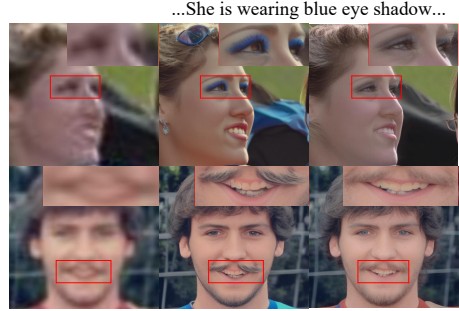
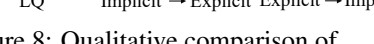

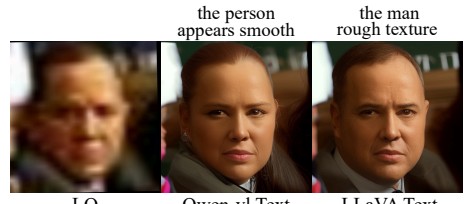

Figure 8: Qualitative comparison of explicit-then-implicit guidance order.

| Dataset | Metrics | Implicit → Explicit | Explicit → Implicit |
|---|---|---|---|
| WIDER-Test | FID ↓ | 36.14 | 35.04 |
| | MANIQA ↑ | 0.54 | 0.57 |
| | CLIPIQA ↑ | 0.72 | 0.74 |
| CelebA-Test (FID=110.05) | FID ↓ | 27.37 | 21.96 |
| | LPIPS ↓ | 0.32 | 0.28 |
| | MANIQA ↑ | 0.60 | 0.59 |
| | CLIPIQA ↑ | 0.73 | 0.71 |
| CelebA-Test (FID=262.42) | FID ↓ | 65.96 | 56.53 |
| | LPIPS ↓ | 0.56 | 0.52 |
| | MANIQA ↑ | 0.57 | 0.57 |
| | CLIPIQA ↑ | 0.72 | 0.70 |

Table 6: Quantitative comparison of the ordering of explicit and implicit textual guidance.

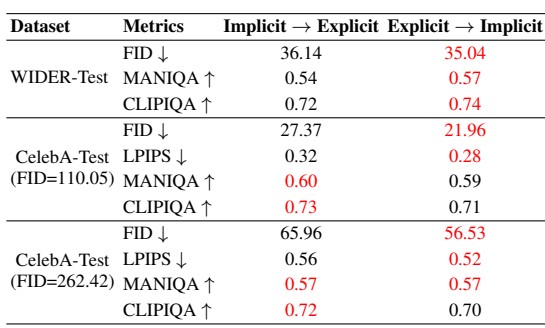

Figure 9: Qualitative comparison of different explicit text extractors.

| Dataset | Metrics | Qwen-vl Text | LLaVA Text |
|---|---|---|---|
| WIDER-Test | FID ↓ | 37.84 | 35.04 |
| | MANIQA ↑ | 0.56 | 0.57 |
| | CLIPIQA ↑ | 0.73 | 0.74 |
| CelebA-Test (FID=262.42) | FID ↓ | 61.24 | 56.53 |
| | LPIPS ↓ | 0.53 | 0.52 |
| | MANIQA ↑ | 0.57 | 0.57 |
| | CLIPIQA ↑ | 0.69 | 0.70 |

Table 7: Quantitative comparison of different explicit text extractors.

**Importance of Guidance Order in DCA.** The order of guidance in DCA is critical. We adopt an explicit-before-implicit sequence, creating a coarse-to-fine workflow. An explicit prompt first establishes a broad semantic anchor, which the implicit guidance then refines to precisely match the ground truth. Inverting this sequence is ineffective, as a broad prompt struggles to correct an already-processed representation, causing over-suppression of details and unnatural textures. As Fig. 8 confirms, this explicit-then-implicit ordering produces more faithful and natural restorations.

**Superiority of LLaVA as an Explicit Text Extractor.** To select our explicit text extractor, we compared LLaVA-v1.5-13B (Liu et al., 2023a) with Qwen2-VL-7B-Instruct (Wang et al., 2024). LLaVA consistently outperforms Qwen2, achieving higher scores on all metrics, especially under severe degradation (Tab. 7). This is due to its superior expressiveness; LLaVA generates detailed, photorealistic captions with attributes like "brown irises" and "crow's feet" that guide the model to more natural results (Fig. 9). Conversely, descriptions from Qwen2 are often generic ("the skin") or inaccurate, sometimes introducing gender ambiguity. For a full textual comparison, see Appendix Sec. D.4.

## 5 CONCLUSION

We proposed **RoboFace**, a novel blind face restoration method that robustly aligns degraded inputs with high-quality reconstructions via complementary guidance. By integrating explicit natural language prompts and implicit CLIP-based features through a Decoupled Cross-Attention module, RoboFace achieves both semantic controllability and faithful reconstruction. Extensive experiments verify its effectiveness, demonstrating exceptional robustness and capacity to handle diverse and out-of-the-distribution degradations.

## 6 ETHICS STATEMENT

This work follows the ICLR Code of Ethics. The datasets used are publicly available and licensed for research; all images are non-identifiable or come from releases that permit research use, and no new data involving human subjects were collected. We took care to avoid harmful or sensitive attributes in prompts and analyses, and report failure cases (e.g., potential attribute drift) to mitigate misuse. No personally identifiable information is included, and all model outputs are intended solely for research. Any potential risks (hallucinated attributes, background artifacts) are discussed in the Limitations section, along with guidance for responsible use.

## 7 REPRODUCIBILITY STATEMENT

To ensure reproducibility, we release an anonymous package with training and inference scripts, exact prompts, model names/versions, checkpoints, and all hyperparameters (optimizer, learning rates, seeds, batch sizes). We also provide data preprocessing steps, evaluation protocols (metrics, masks/regions), and environment specifications (CUDA/driver, PyTorch/diffusers/transformers versions). Our code will be submitted in the supplementary material.

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

# A STATEMENT OF THE USE OF LARGE LANGUAGE MODELS (LLMs)

During our experiments, we used vision–language models to derive textual descriptions and to quantify textual similarity. Specifically, LLaVA-v1.5-13B and Qwen2-VL-7B-Instruct were employed to generate image-level textual descriptions, and Qwen3-Embedding-6B was used to compute semantic similarity between captions.

Consistent with the ICLR 2026 policy on LLM usage, we used LLMs solely as assistive tools for light editorial polishing (grammar, wording, and formatting) of author-written text. LLMs did not contribute to research ideation, experimental design, analysis, or substantive writing, and are not eligible for authorship.

# B MORE RESULTS ON RoboFace

## B.1 MORE RESULTS ON BLIND FACE RESTORATION.

In this section, we provide more visual comparisons with state-of-the-art methods, including GFP-GAN (Wang et al., 2021a), GPEN (Yang et al., 2021), CodeFormer (Zhou et al., 2022), DAEFR Tsai et al. (2024), DR2 (Wang et al., 2023b), DifFace (Yue & Loy, 2024), DiffBIR (Lin et al., 2024b).

**More results on SCface dataset(Extreme real surveillance degradations).** As shown in Fig. 10, our method effectively handles unseen noise in real-world surveillance degraded images, restoring more realistic and natural facial features while maintaining consistency with the gallery photos.

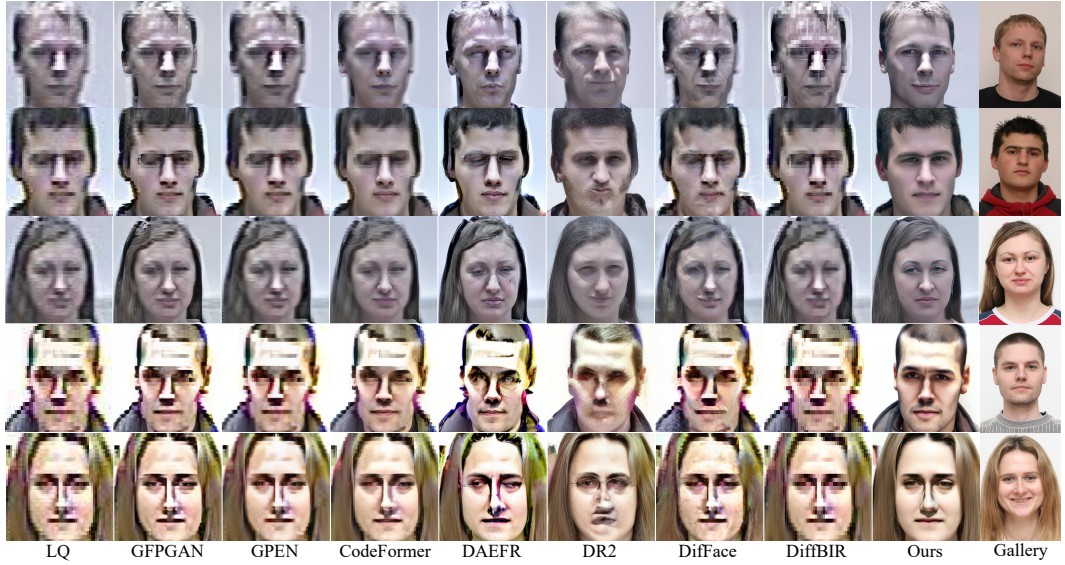

LQ        GFPGAN      GPEN      CodeFormer      DAEFR      DR2      DifFace      DiffBIR      Ours      Gallery

Figure 10: Visual comparison of different methods on SCface dataset.

**More Results on CelebA-Test under varying degradations.** We present additional visual comparisons with state-of-the-art methods on CelebA-Test to highlight the robustness of our approach. Under out-of-distribution noise (FID = 262.42), our method recognizes unseen noise patterns, indicating strong generalization of the implicit textual guidance and the representational capacity of the textual space. Despite complex corruptions, it still produces faces with similar identities and fine-grained textures as shown in Fig. 11.

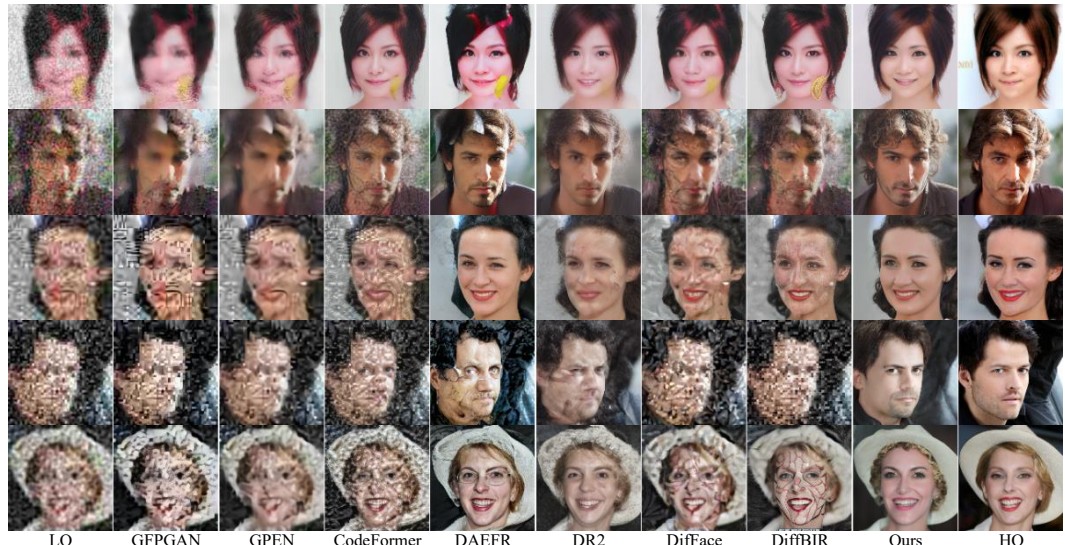

LQ    GFPGAN    GPEN    CodeFormer    DAEFR    DR2    DifFace    DiffBIR    Ours    HQ

Figure 11: Visual comparison of different methods on CelebA-Test(FID=262.42) dataset.

## B.2 MORE RESULTS ON IMPLICIT TEXT EXTRACTOR.

As shown in Fig. 12, our Mapper trained with unpaired data demonstrates strong reconstruction ability: given HQ inputs, it produces semantically consistent, high-fidelity outputs, effectively encoding facial content and identity. As illustrated in Fig. 13, the Mapper outputs for LQ inputs indicate that the Mapper learns degradation characteristics well.It also shows that our Remover, trained with paired LQ-HQ data, can generate high-quality results from degraded inputs. It effectively removes various distortions and produces natural-looking faces, thereby providing reliable implicit textual features.

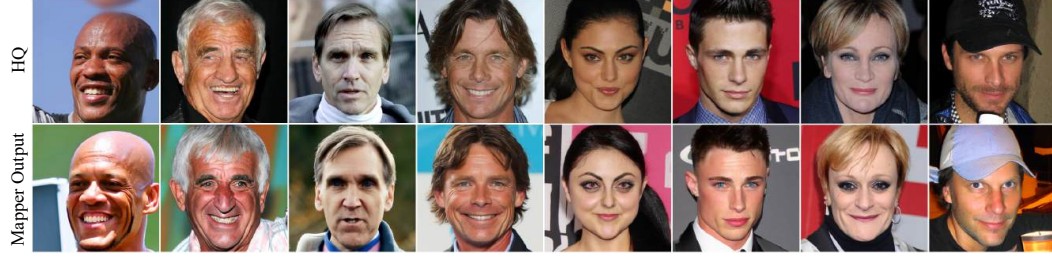

Figure 12: Visual results of Mapper output (HQ) on the CelebA-Test dataset.

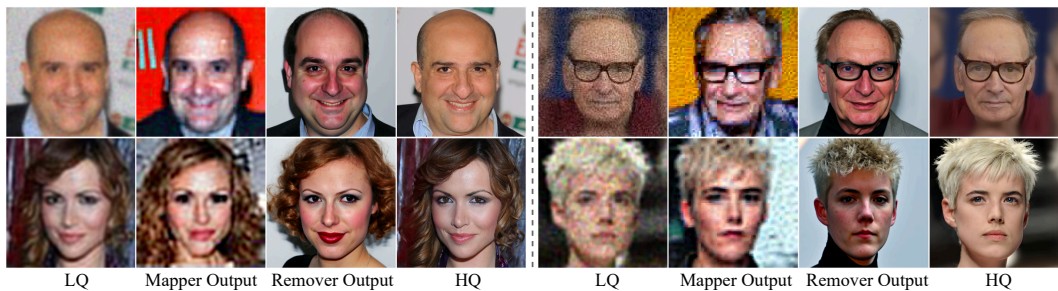

LQ    Mapper Output  Remover Output    HQ        LQ    Mapper Output  Remover Output    HQ

Figure 13: Visual results of Remover output on the CelebA-Test dataset.

# C    MORE DETAILS OF ROBOFACE

## C.1    MORE DETAILS OF THE ROBOFACE DEGRADATION PIPELINE.

We train all models on the FFHQ dataset (Karras et al., 2019), which contains 70,000 high-quality face images resized to $512 \times 512$. To generate realistic LQ training inputs, we follow the second-order degradation strategy proposed in Real-ESRGAN (Wang et al., 2021b). Each HQ image ($y$) undergoes two sequential degradation processes, and the overall degradation process in one stage can be formulated as:

$$x = [(y \otimes k_\sigma) \downarrow_r + n_\delta]_{\mathrm{JPEG}_q}, \tag{5}$$

where $x$ is the resulting low-quality (LQ) image. The degradation includes Gaussian blur ($k_\sigma$), downsampling by a factor $r$ ($\downarrow_r$), additive Gaussian noise $n_\delta$, and JPEG compression with quality factor $q$ ($\mathrm{JPEG}_q$). Each degradation stage shares this structure but uses independently sampled parameters. Specifically, following DiffBIR (Lin et al., 2024b), we randomly sample the degradation parameters $\sigma$, $r$, $\delta$, and $q$ from the ranges $[0.2, 3]$, $[0.15, 1.5]$, $[1, 30]$, and $[30, 95]$ in the first degradation stage, and $[0.2, 1.5]$, $[0.3, 1.2]$, $[1, 25]$, and $[30, 95]$ in the second stage, respectively.

## C.2    MORE DETAILS OF THE DECOUPLED CROSS ATTENTION MECHANISMS IN CONTROLNET.

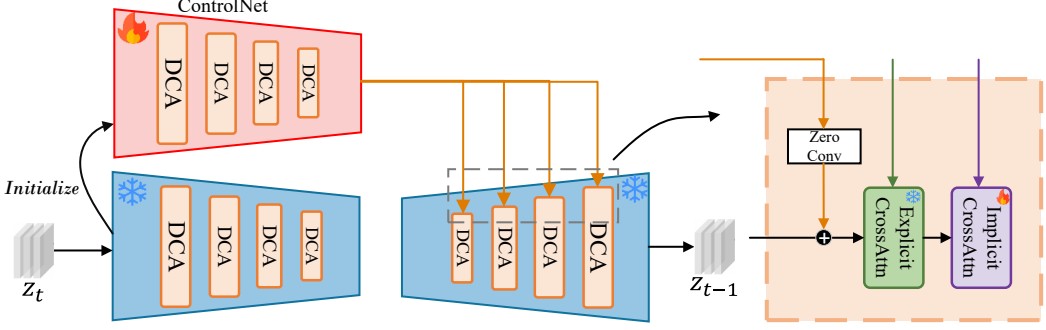

Figure 14: Overview of the integration between ControlNet and the pre-trained Stable Diffusion.

By embedding DCA blocks within each layer of the ControlNet, we ensure that the conditioning process is guided by both explicit and implicit textual streams, which also guide the primary denoising process. This parallel guidance mechanism enables the ControlNet to learn robust, degradation-aware features that remain semantically aligned with the final restoration target, fostering a coherent and synergistic interaction between the learned conditions and the generative prior.

# D MORE DISCUSSIONS ON ROBOFACE

## D.1 NECESSITY OF REMOVER IN THE IMPLICIT TEXT EXTRACTOR.

As shown in Tab. 8, incorporating the proposed remover consistently improves performance across all datasets. On WIDER-Test and both levels of CelebA-Test degradations, the results with remover achieve the best scores in all metrics. This gain comes from the remover's ability to eliminate noise in the textual representation, yielding cleaner and more faithful identity cues. By projecting these refined signals into a more accurate structured semantic space, the model can achieve higher-fidelity restorations and demonstrate stronger robustness under complex degradations.

| Metric | WIDER-Test | | CelebA-Test (FID=110.50) | | CelebA-Test (FID=262.42) | |
|---|---|---|---|---|---|---|
| | w/o remover | w/ remover | w/o remover | w/ remover | w/o remover | w/ remover |
| FID ↓ | 37.67 | **35.04** | 24.46 | **21.96** | 68.46 | **56.53** |
| LPIPS ↓ | - | - | 0.2897 | **0.2795** | 0.5382 | **0.5239** |
| MANIQA ↑ | 0.5502 | **0.5664** | 0.5658 | **0.5934** | 0.5367 | **0.5708** |
| CLIP-IQA ↑ | 0.7182 | **0.7362** | 0.6749 | **0.7068** | 0.6562 | **0.6973** |

Table 8: Ablation results show the Necessity of remover in Implicit Text Extractor.

## D.2 COMPARISON OF TOKEN NUMS IN MAPPER AND REMOVER.

We vary the number of textual tokens $N$ used by the mapper and remover to control representational capacity. When $N = 20$, the representation capacity is insufficient to capture rich semantic concepts, leading to degraded performance. When $N = 40$, the enlarged input space requires the model to process more word tokens, which may introduce irrelevant information and additional noise or inconsistent representations, thereby deteriorating the restoration quality. As a result, $N = 30$ strikes a good balance between representation capacity and robustness, consistently achieving the best performance across metrics.

| Metric | WIDER-Test | | | CelebA-Test (FID=110.05) | | | CelebA-Test (FID=262.34) | | |
|---|---|---|---|---|---|---|---|---|---|
| Token Numbers | 20 | 30 | 40 | 20 | 30 | 40 | 20 | 30 | 40 |
| FID ↓ | 38.17 | **35.04** | 37.91 | 23.29 | **21.96** | 25.70 | 66.93 | **56.53** | 67.71 |
| LPIPS ↓ | - | - | - | 0.2868 | **0.2795** | 0.2901 | 0.5325 | **0.5239** | 0.5394 |
| MANIQA ↑ | 0.5538 | **0.5664** | 0.5460 | 0.5704 | **0.5934** | 0.5607 | 0.5371 | **0.5708** | 0.5307 |
| CLIP-IQA ↑ | 0.7254 | **0.7362** | 0.7146 | 0.6917 | **0.7050** | 0.6813 | 0.6596 | **0.6973** | 0.6598 |

Table 9: Comparison of token numbers in Mapper and Remover on CelebA-Test and WIDER-Test.

## D.3 EFFECT OF PROMPT DESIGN.

The performance of LLaVA (Liu et al., 2023a) is highly dependent on well-crafted prompts (Liu et al., 2023b). We use the prompt: *Provide a detailed yet concise description of this person's face. Include their face shape, eyes, nose, mouth, eyebrows, skin texture and tone, expression, and any notable features like moles, freckles, or wrinkles.* to focus the output on detailed facial component structures. We show LLaVA outputs under three prompt types. As illustrated in Fig. 15, facial description prompts yield more face-specific details.

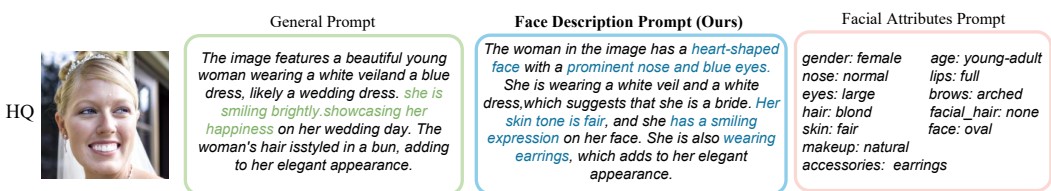

Figure 15: Comparison of general prompt, face description prompt and facial attributes prompt.

### D.4 COMPARISON OF DETAILED TEXT DESCRIPTIONS FROM DIFFERENT TEXT EXTRACTORS: LLAVA, QWEN-VL

We use the same query inputs as in Appendix Sec. D.3 and apply them to LLaVA-v1.5-13B (Liu et al., 2023a) and Qwen2-VL-7B-Instruct (Wang et al., 2024) to extract the corresponding text descriptions. As shown in Fig. 16, Qwen-vl tends to produce more generic and template-like descriptions, often overlooking salient details. For example, it labels the face as "round" and the expression as "neutral," while also ignoring fine-grained traits such as the eye squint or the distinctive shape of the mouth. In contrast, LLaVA provides more specific and realistic details, capturing attributes like a "square-shaped face," "wide-set eyes with a noticeable squint," and "thick, arched eyebrows." These richer cues lead to captions that are more faithful to the actual visual content, thereby offering better semantic grounding for downstream restoration.

| HQ | Qwen-vl Text | LLaVA Text |
|---|---|---|
| | *The **person** in the image has **a round face shape**. Their eyes are **light-colored and positioned slightly above the center of their face**. The nose is straight and prominent. The mouth is closed and has a neutral expression. The eyebrows are **thin and arch slightly**. The skin texture **appears smooth with a light tone**. There are no visible moles, freckles, or wrinkles. The background is a solid blue color.* | *The boy in the image has **a square-shaped face with a prominent nose and a thin mouth**. His eyes are **wide-set and have a noticeable squint**, giving him **a somewhat quizzical expression**. His eyebrows are **thick and arched**, adding to his **distinctive appearance**. The boy's skin has a light tone and a smooth texture, with no visible moles, freckles, or wrinkles. Overall, his face is characterized by a **combination of distinctive features that make him stand out**.* |

Figure 16: Comparison of detailed text descriptions from different text extractors.

### D.5 EXPLICIT TEXTUAL GUIDANCE FOR CONTROLLABLE RESTORATION

Our framework supports explicit guidance from natural language text prompts, enabling flexible and user-interactive control. As illustrated in Fig. 17, users can manipulate attributes such as eyes, hair, mustache, gender and so on. When the degradation level of the LQ image is relatively mild, the influence of explicit text guidance becomes weaker, and the restoration tends to prioritize fidelity to the original degraded input. In contrast, under severe degradations where fine details are no longer discernible by the human eye, textual prompts play a more significant role by providing controllable cues to guide the generation of otherwise missing attributes.

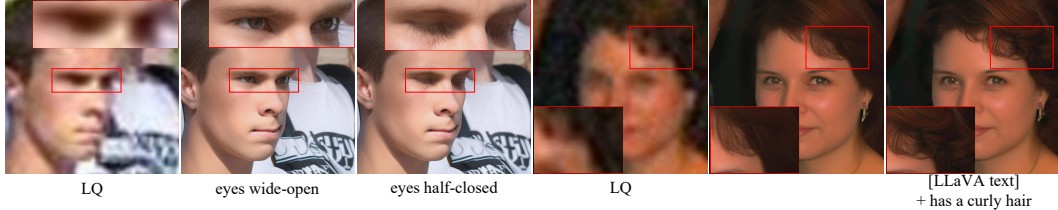

| LQ | eyes wide-open | eyes half-closed | LQ | | [LLaVA text] + has a curly hair |

Figure 17: Qualitative examples of explicit textual guidance for controllable face restoration.

### D.6 COMPARISON OF MODEL COMPLEXITY AND PERFORMANCE.

We compare RoboFace against state-of the-art diffusion-based methods. As shown in Tab. 10, while our model introduces a moderate level of complexity, it consistenty achieves the best overall performance, as demonstrated by both quantitative metrics and qualitative visual results (see Fig. 4 and Tab. 3). Furthermore,it requires only two stages of tuning and does not rely on hyperparameter adjustment, making it relatively efficient in practice.

| Methods | Params (M) | FLOPs (G) | Inference Time |
|---|---|---|---|
| DiffFace | 175.40 | 185.95 | 7.05s |
| DR2 | 85.79 | 725.08 | 2.05s |
| DiffBIR | 1716.7 | 900.52 | 9.03s |
| **RoboFace** | 2590.5 | 704.48 | 4.32s |

Table 10: Comparison of model complexity and performance.

# E   LIMITATIONS

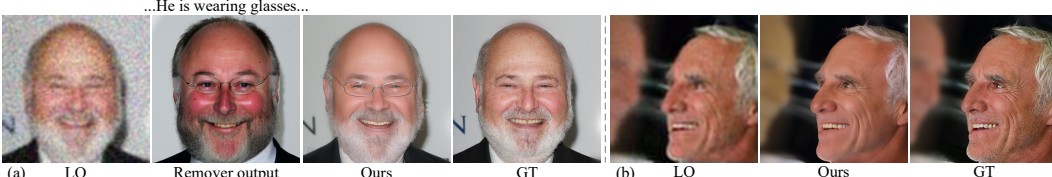

Figure 18: Examples of limitations. (a) captions from remover outputs may include inaccurate attributes (e.g., glasses). (b) restored results may show background inconsistency.

Despite the promising performance of RoboFace, several limitations remain. First, captions are obtained from HQ images during training (FFHQ) and from near-HQ remover outputs at inference rather than directly from severely degraded LQ inputs; remover references can lack fine facial details, making captions less discriminative or occasionally inaccurate, which weakens guidance and may cause mild attribute drift (see Fig. 18(a)). Second, because our framework builds on a pretrained Stable Diffusion model, it inherits the tendency of diffusion methods to prioritize semantic and structural fidelity over background consistency, sometimes yielding slight color shifts or artifacts in the background (see Fig. 18(b)).

