# OpenReview forum: "RoboFace: Face Restoration Made Robust via Implicit and Explicit Textual Guidance"
_ICLR.cc/2026/Conference — ICLR 2026 Conference Withdrawn Submission_

### Official Review · Reviewer_RDH7 · 2025-10-31

**Soundness:** 2
**Presentation:** 2
**Contribution:** 2
**Rating:** 4
**Confidence:** 4

**Summary:**

This paper proposes RoboFace, a diffusion-based framework for face restoration under real-world degradations. The method introduces a dual textual guidance mechanism consisting of (1) implicit guidance, where CLIP image embeddings are mapped into the text space and denoised via a Remover network, and (2) explicit guidance, where natural-language descriptions generated by LLaVA are injected via a decoupled cross-attention (DCA) design. The goal is to enhance both robustness and controllability while preserving identity consistency. Experiments on multiple benchmarks demonstrate strong visual quality and robustness to out-of-distribution (OOD) degradations.

**Strengths:**

1. Combining explicit (language-driven) and implicit (CLIP-driven) semantic signals in a decoupled attention order (“explicit → implicit”) is a technical contribution.

2. The method is clearly explained, and the system pipeline (dual textual paths + DCA) is easy to follow.

**Weaknesses:**

1. I am quite confused why  CLIP-based priors is treated as implicit condition. The text encoder of clip can explicitly encode semantic information of the input text. I think it is similar to direct text input.

2. The core components (CLIP-based priors, text conditioning, and correlation-aware loss) have already been explored in recent studies such as DiffBIR, DifFace, and correlation-regularized diffusion approaches.

3. The manuscript does not sufficiently position its contribution against newer 2025 methods like LAFR (2025), TD-BFR (2025), HonestFace (2025), OSDFace (2024), and DiffusionReward (2025), which report stronger fidelity, efficiency, and identity consistency.

4. The experiments omit several very recent, high-performing diffusion-based face restoration methods (e.g., LAFR, TD-BFR, HonestFace, DiffusionReward, VSPBFR). Without these, it is hard to assess the relative advantage of RoboFace, especially in terms of efficiency and identity preservation.

5. The performance may be affected by LLaVA’s description quality under extreme degradations. The paper briefly notes this but does not analyze failure cases or ablations on prompt noise.

**Questions:**

See weakness

---

### Official Review · Reviewer_dLaE · 2025-10-31

**Soundness:** 2
**Presentation:** 2
**Contribution:** 2
**Rating:** 2
**Confidence:** 5

**Summary:**

The authors focus on the problem of face restoration and propose a novel blind face restoration framework that leverages dual textual guidance to enhance robustness against diverse degradations.  Specifically, the proposed method integrates implicit guidance from CLIP latent features and explicit guidance from natural language prompts through a Decoupled Cross-Attention (DCA) module, ensuring high-fidelity and user-controllable restorations. Extensive experiments show superior performance over state-of-the-art methods, particularly in handling severe and out-of-distribution degradations.

**Strengths:**

1. This work proposes a dual textual guidance framework that leverages implicit CLIP features and explicit natural-language prompts, using semantic space as an intermediate representation to enable blind face restoration without explicit degradation modeling.
2. A Decoupled Cross-Attention (DCA) module with an explicit-then-implicit fusion order is introduced to preserve diffusion priors while enhancing identity consistency.

**Weaknesses:**

1. LLaVA-generated descriptions may misidentify facial attributes under severe degradation, and such errors are injected as explicit guidance into the diffusion model, potentially causing identity drift.
2. RoboFace has a large model size (2590M parameters) and inference latency (4.32s), limiting its practicality in resource-constrained settings.
3. Explicit prompts rely on an intermediate restoration from Stage I as input to LLaVA, introducing a two-stage pipeline that increases complexity and hinders end-to-end deployment, as final quality depends heavily on the intermediate result.

**Questions:**

1. Have the authors tried generating explicit text directly from the raw LQ input? If not feasible, could LLaVA outputs be post-processed or equipped with a confidence mechanism to mitigate erroneous guidance?
2. The DCA module uses an explicit-then-implicit fusion order, but the rationale for not using parallel or implicit-then-explicit fusion is unclear. While Table 6 shows performance differences, the underlying mechanism lacks analysis.
3. As shown in Table 10, RoboFace has 2590M parameters and 4.32s inference time. Can the architecture be simplified for deployment? Does the current design strike a reasonable trade-off between speed and quality?

---

### Official Review · Reviewer_TMN5 · 2025-10-31

**Soundness:** 3
**Presentation:** 3
**Contribution:** 2
**Rating:** 2
**Confidence:** 3

**Summary:**

This paper presents RoboFace, a novel blind face restoration (BFR) framework that leverages dual textual guidance (implicit CLIP features and explicit natural language prompts) to address out-of-distribution (OOD) degradations. The approach is innovative, well-motivated, and demonstrates strong empirical results.

**Strengths:**

Robustness: The method generalizes exceptionally well to OOD degradations, as validated on synthetic (CelebA-Test) and real-world (SCface, LFW, WIDER) datasets. The explicit-then-implicit guidance order is justified through ablations.

Rigorous Evaluation: Extensive comparisons with 7 SOTA methods across multiple degradation levels and metrics (FID, LPIPS, CLIPIQA) convincingly demonstrate superiority. The t-SNE analysis and text-similarity experiments provide insightful validation of the proposed textual-space denoising.

Controllability: The framework supports user-interactive restoration via explicit text prompts (e.g., modifying facial attributes), adding practical value.

**Weaknesses:**

Questionable Validity of OOD Problem Formulation: The paper's central argument—that existing BFR methods struggle with out-of-distribution (OOD) degradations—relies heavily on evaluating low-quality face cases using FID scores. However, the experimental setup may overstate the practical relevance of the OOD scenario. For instance, the last row of Figure 3 shows that the so-called "LQ faces" are almost pure noise, which shifts the problem from faithful restoration to noise-to-image generation. In such extreme cases, it is inherently difficult to guarantee the authenticity of reconstructed facial identities, potentially undermining the core objective of BFR. This raises doubts about whether the proposed OOD benchmark aligns with real-world restoration needs.

Insufficient Comparison with Contemporary Diffusion-Based Methods: While the authors compare RoboFace with several established methods (e.g., CodeFormer, GFPGAN), the literature review overlooks recent advances in diffusion-based BFR. For a fair and comprehensive evaluation, the following works should be discussed and compared:

DR2(Wang et al., 2023): A diffusion-based degradation remover designed for robust blind face restoration.

DiffBIR(Lin et al., 2024): Leverages generative diffusion priors for real-world blind image restoration.

AuthFace(Chen et al., 2025): Focuses on authentic face restoration with face-oriented diffusion priors.

**Questions:**

Please refer to the paper weakness

---

### Official Review · Reviewer_yqMg · 2025-11-04

**Soundness:** 3
**Presentation:** 3
**Contribution:** 3
**Rating:** 6
**Confidence:** 4

**Summary:**

The paper proposes RoboFace, a blind face restoration (BFR) framework that couples CLIP-derived features mapped into the text-conditioning space—and explicit textual guidance—natural-language prompts—inside a Decoupled Cross-Attention (DCA) conditioning scheme for a pretrained diffusion model. The approach aims to improve robustness to out-of-distribution (OOD) degradations without retraining a large diffusion prior. Experiments on synthetic CelebA splits (in-range, partially OOD, fully OOD) and real datasets (LFW-Test, WIDER-Test, SCface) show strong perceptual quality and improved FID on challenging settings.

**Strengths:**

Implicit CLIP-to-text tokens (Mapper→Remover) plus explicit prompts, fused via explicit-then-implicit DCA to first set global semantics and then refine identity—a coherent coarse-to-fine logic. On the hardest CelebA split (FID≈262), RoboFace attains the best overall perceptual metrics and competitive FID vs. diffusion and codebook baselines; on SCface, it outperforms alternatives on average FID and maintains high CLIPIQA. Studies dissect effects of (i) implicit vs. explicit guidance, (ii) DCA ordering, (iii) text extractor choice (LLaVA vs. Qwen-VL), and (iv) token count, supporting design choices.

**Weaknesses:**

1. During inference, explicit prompts come from LLaVA run on an intermediate restoration (Stage I) rather than the LQ image; errors or biases here may propagate. The paper acknowledges occasional attribute drift.
2. Training uses second-order degradations; benchmarks focus on blur/noise/JPEG and surveillance. Fewer results on other real pipelines (e.g., heavy compression artifacts beyond JPEG ranges, motion blur, color shifts).
3. Model size is large (due to SD 2.1 + ControlNet + dual guidance); while inference time is reported as reasonable, parameters/FLOPs are substantially higher than some baselines. A stronger efficiency-quality tradeoff analysis would help.

**Questions:**

1. How sensitive is performance to LLaVA prompt errors? Can you quantify restoration quality as you inject noise or replace LLaVA with weaker captions?
2. What is the gap between explicit prompts generated from the Stage-I image vs. directly from the LQ input, especially on SCface?
3. Can you include tests on motion blur, non-Gaussian noise, color/band-limited artifacts, or smartphone pipeline degradations

---

### Note · Authors · 2025-11-13

I have read and agree with the venue's withdrawal policy on behalf of myself and my co-authors.